
# Statistical post-processing of ensemble forecasts of the height of new snow

Jari-Pekka Nousu[1,2], Matthieu Lafaysse[2], Matthieu Vernay[2], Joseph Bellier[3,4], Guillaume Evin[5], and Bruno Joly[6]

[1]University of Oulu, Water, Energy and Environmental Engineering Research Unit, Oulu, Finland
[2]Univ. Grenoble Alpes- Université de Toulouse- Météo-France- CNRS- CNRM, Centre d'Etudes de la Neige, Grenoble, France
[3]Cooperative Institute for Research in Environmental Sciences- University of Colorado Boulder- and NOAA Earth System Research Laboratory, Physical Sciences Division, Boulder- Colorado, USA
[4]Univ. Grenoble Alpes, CNRS, IRD, Grenoble INP, IGE, Grenoble, France
[5]Univ. Grenoble Alpes - IRSTEA, UR ETNA, Grenoble, France
[6]CNRM- Université de Toulouse- Météo-France- CNRS, GMAP, Toulouse, France

**Correspondence:** matthieu.lafaysse@meteo.fr

**Abstract.**

Forecasting the height of new snow (HN) is crucial for avalanche hazard forecasting, roads viability, ski resorts management and tourism attractiveness. Meteo-France operates the PEARP-S2M probabilistic forecasting system including 35 members of the PEARP Numerical Weather Prediction system, where the SAFRAN downscaling tool is refining the elevation resolution, and the Crocus snowpack model is representing the main physical processes in the snowpack. It provides better HN forecasts than direct NWP diagnostics but exhibits significant biases and underdispersion. We applied a statistical post-processing to these ensemble forecasts, based on Nonhomogeneous Regression with a censored shifted Gamma distribution. Observations come from manual measurements of 24-hour HN in French Alps and Pyrenees. The calibration is tested at the station-scale and the massif-scale (i.e. aggregating different stations over areas of 1000 km$^2$). Compared to the raw forecasts, similar improvements are obtained for both spatial scales. Therefore, the post-processing can be applied at any point of the massifs. Two training datasets are tested: (1) a 22-year homogeneous reforecast for which the NWP model resolution and physical options are identical to the operational system but without the same initial perturbations; (2) 3-year real-time forecasts with a heterogeneous model configuration but the same perturbation methods. The impact of the training dataset depends on lead time and on the evaluation criteria. The long-term reforecast improves the reliability of severe snowfall but leads to overdispersion due to the discrepancy in real-time perturbations. Thus, the development of reliable automatic forecasting products of HN needs long reforecasts as homogeneous as possible with the operational systems.

## 1 Introduction

Forecasting the height of new snow (HN, Fierz et al., 2009) is essential in the mountainous areas as well as in the northern regions due to various safety issues and economic activities. For instance, avalanche hazard forecasting, roads viability, ski



resort management and tourism attractiveness rely on the forecasts of HN. Automatic predictions are increasingly developed for that purpose, based on Numerical Weather Prediction (NWP) models output. Nevertheless, accurate forecasting of this variable is still challenging for several reasons. First, the precipitation forecasts in NWP models have significant errors which increase with longer lead times. These forecast uncertainties have to be considered. Second, the high variability of HN as a

function of elevation is difficult to describe in mountainous areas, even at the best spatial resolution available in NWP models (i.e. 1 or a few km). Finally, several processes such as density of falling snow, mechanical compaction during the deposition and variations of the rain-snow limit elevation during some storm events are not or poorly represented in NWP models. Several recent scientific advances can help to face these challenges:

  – To estimate forecast uncertainty, ensemble forecasting has become an important method in NWP. Probabilistic forecasts
have been in operational use for number of years in several meteorological centres (Molteni et al., 1996; Toth and Kalnay, 1997; Pellerin et al., 2003). Ensemble forecasting has increased the confidence of forecast users to predict possible future occurrence, or non-occurrence, of unusually strong events (Candille and Talagrand, 2005). In many cases, an estimation of the probability density of future weather-related variable may present more value for the forecast user than a single deterministic forecast does (Richardson, 2000; Ramos et al., 2013). The forecast uncertainties depend on the atmospheric
flow and vary from day to day (Leutbecher and Palmer, 2008). Therefore, ensemble forecasting aims at estimating the probability density of the future state of the atmosphere.

  – In NWP, snowpack modelling is necessary since the presence of snow on the ground has major impact on all the fluxes taking place at the interface between Earth's atmosphere and its surface. However, NWP models often use single-layer snow schemes with homogeneous physical properties because they are relatively inexpensive, have relatively few param-
eters and capture first order processes (Douville et al., 1995). Models with more complexity have also been developed but are not yet implemented in most NWP systems. The most detailed ones are able to represent a detailed stratigraphy of the snowpack with an explicit description of the time evolution of the snow microstructure (Lehning et al., 2002; Vionnet et al., 2012). Snow Model Intercomparison Projects (Krinner et al., 2018) suggest that detailed snowpack models are among the most accurate models in the reproduction of the snowpack evolution in various climates and environments.
Operationally, these snow models are sometimes forced by NWP outputs to forecast the risk of avalanche (Durand et al., 1999). Concerning the topic of this study, it is known that these models also provide better estimates of the height of new snow than direct NWP outputs (Champavier et al., 2018). This is explained by the abililty of these schemes to simulate the mechanical compaction of snow on the ground occurring during the snowfall, the possible impact of changes in pre-cipitation phase during a storm event, the possible occurrence of melting at the surface or at the bottom of the snowpack
and the dependence of falling snow density on meteorological conditions.

    To benefit from both the advantages of ensemble NWP and detailed snowpack modelling, Vernay et al. (2015) devel-oped the PEARP-S2M modelling system (PEARP: Prévision d'Ensemble ARPEGE; ARPEGE: Action de Recherche Petite Echelle Grande Echelle; S2M: SAFRAN-SURFEX-MEPRA; SAFRAN: Système Atmosphérique Fournissant des Renseigne-ments Atmosphériques à la Neige; SURFEX: SURFace EXternalisée; MEPRA: Modèle Expert pour la Prévision du Risque



d'Avalanches). In this system, the Crocus detailed snowpack model (Vionnet et al., 2012) implemented in the SURFEX surface modelling platform is forced by the ensemble version of the ARPEGE NWP model (Descamps et al., 2014), after an elevation-adjustment of the meteorological fields by the SAFRAN downscaling tool (Durand et al., 1998). However, the PEARP-S2M system still suffers from various biases and deficiencies (Vernay et al., 2015; Champavier et al., 2018). Biases in atmospheric

ensemble forecasts may be caused by insufficient model resolutions (Weisman et al., 1997; Mullen and Buizza, 2002; Szunyogh and Toth, 2002; Buizza et al., 2003), suboptimal physical parameterizations (Palmer, 2001; Wilks, 2005) or suboptimal methods for generating the initial conditions (Barkmeijer et al., 1998, 1999; Hamill et al., 2000, 2003; Sutton et al., 2006). In case of HN forecasts, the errors origin also from the snow models (Essery et al., 2013; Lafaysse et al., 2017). Due to the systematic biases in ensemble forecasts and the challenge of detecting and correcting their origins, many methods of statistical

post-processing have been developed that leverage archives of past forecast errors (Vannitsem et al., 2018). In the literature, these probabilistic post-processing methods are often refered as Ensemble Model Output Statistics (EMOS) as an extension to ensemble approaches of the traditional Model Output Statistics (MOS) applied for several decades to deterministic forecasts (Glahn and Lowry, 1972). EMOS are now routinely applied for meteorological predictands such as temperature, precipitation and wind-speed. The techniques are for instance nonhomogeneous regression methods (Jewson et al., 2004; Gneiting

et al., 2005; Wilks and Hamill, 2007; Thorarinsdottir and Gneiting, 2010; Lerch and Thorarinsdottir, 2013; Scheuerer, 2014; Scheuerer and Hamill, 2015; Thorarinsdottir and Gneiting, 2010; Baran and Nemoda, 2016; Gebetsberger et al., 2017), logistic regression methods (Hamill et al., 2004; Hamill and Whitaker, 2006; Messner et al., 2014), Bayesian model averaging (Raftery et al., 2005), rank histogram recalibration (Hamill and Colucci, 1997), ensemble dressing approaches (i.e., kernel density) (Roulston and Smith, 2002; Wang and Bishop, 2005; Fortin et al., 2006), and quantile regression forests (Taillardat et al., 2016,

20 2019).

However, statistical post-processing of ensemble HN forecasts is rarely reviewed in the literature. Stauffer et al. (2018) and Scheuerer and Hamill (2019) are the first studies to the best of our knowldege to present post-processed ensemble forecasts of HN. However, they only considered direct ensemble NWP output as predictors (precipitation and temperature) and did not incorporate physical modelling of the snowpack. It can be expected that physical modelling could capture some complex

features explaining the variability of HN. This variability is difficult to reach by multivariate statistical relationships, especially the common high temporal variations of temperature and precipitation intensity during a storm event with highly non-linear impacts on the height of new snow. Furthermore, because they do not consider direct predictors of HN, these recent studies partly rely on precipitation observations in their calibration procedure whereas solid precipitation are particularly prone to very high measurement errors (Kochendorfer et al., 2017). The physical simulation of HN enables considering directly observations

of this variable for the postprocessing. This is a major advantage because HN measurement errors (typically 0.5 cm, WMO, 2018) are considerably lower than errors in solid precipitation measurements.

The goal of this study is to test the ability of a nonhomogeneous regression method to improve the ensemble forecasts of HN from the PEARP-S2M ensemble snowpack modelling system. More precisely, the regression method of Scheuerer and Hamill (2015) based on the Censored Shifted Gamma Distribution was chosen in this work for the advantages identified by

the authors in the case of precipitation forecasts. In particular, this method allows to extrapolate the statistical relationship





between predictors and predictands from common events to more unusual events. Considering the specificities of the available datasets in terms of predictands and predictors, two other scientific questions are considered: (1) can statistical postprocessing be applied at a larger spatial scale than the observation points? (2) what are the requirements of a robust training forecast dataset for statistical postprocessing?

The structure of the paper is as follows. Section 2 describes the model components of the PEARP-S2M system, the observation and forecast datasets used in this study, the nonhomogeneous regression method chosen for post-processing and the evaluation metrics. In Sect. 3, the results of the post-processing method are presented for different training configurations. The discussion in Sect. 4 focuses on the implications of our study about the possibility to implement such post-processing in operational automatic forecast products and recommendations for improvements.

## 10  2   Data and methods

### 2.1   Models

#### 2.1.1   PEARP ensemble NWP system

PEARP is a short-range ensemble prediction system operated by Météo-France up to 4.5 days, fully described in Descamps et al. (2014). It includes 35 forecast members of the ARPEGE NWP model. In 2019, it is based on a 25-member ensemble
assimilation combined with the singular vectors perturbations methods (Buizza and Palmer, 1995; Molteni et al., 1996) to provide 35 initial states. The singular vector perturbations are designed to optimize the spread of the large-scale atmospheric fields at a 24h lead time. Finally, the 35 members are randomly associated with 10 different sets of physical parameterizations including different deep and shallow convection schemes, among others. The current horizontal resolution is about 10 km over France (truncature T798C2.4) with 90 atmospheric levels. All these features have been improved over time with a new
operational configuration provided almost every year.

#### 2.1.2   SAFRAN downscaling tool

SAFRAN (Durand et al., 1993, 1998) is a downscaling and surface analysis tool specifically designed to provide meteorological fields in moutainous areas (i.e. with high elevation gradients). The principle of SAFRAN is to perform a spatialization of the available weather data in mountain ranges so-called "massifs" of about 1000 km$^2$ where meteorological conditions are assumed
to depend only on altitude. SAFRAN variables include precipitation (rainfall and snowfall rate), air temperature, relative humidity, wind speed as well as incoming longwave and shortwave radiations. Although SAFRAN was initially designed to work as an analysis system adjusting a guess from NWP outputs with the available meteorological observations, SAFRAN also comes with a forecast mode which can be considered simply as a downscaling tool to convert NWP model grid (PEARP in our case) to the massif geometry. The originality of this system is the use of different vertical levels of the NWP model to
obtain surface fields at different elevations.





### 2.1.3 Crocus snowpack model

Crocus (Vionnet et al., 2012) is a one-dimensional multilayer physical snow scheme which simulates the evolution of the snow cover affected by both the atmosphere and the ground below. It is implemented in the SURFEX surface modelling platform (Masson et al., 2013) as the most detailed snow scheme of the ISBA land surface model. Each snow layer is described by

its mass, density, enthalpy (temperature and liquid water content) and age. The evolution of snow grains is described with additional variables (optical diameter and sphericity) using metamorphism laws from Brun et al. (1992) and Carmagnola et al. (2014). Snow density is a particularly important property for the height of new snow. It is mainly affected by two key processes: the density of falling snow and the compaction of snow on the ground. Falling snow density was empirically parameterized as a function of air temperature and wind speed (Pahaut, 1975). This parameterization is associated with significant uncer-

tainties (Lafaysse et al., 2017; Helfricht et al., 2018). Snow compaction is modelled with a visco-elastic scheme in which the snow viscosity of each layer is parameterized depending mainly on the layer density and temperature. The parameterization of snow viscosity is also uncertain as various expressions were formulated in the literature (Teufelsbauer, 2011). Furthermore, the compaction velocity actually has a high dependence to snow microstructure (Lehning et al., 2002). This complex dependence cannot be described in Crocus by the visco-elastic concept and microstructure-dependent models of compaction are only avail-

able for very specific conditions (Schleef et al., 2014). These limitations partly explain the errors of simulated HN identified by Champavier et al. (2018) in combination with the known errors and underdispersion of precipitation input.

## 2.2   Data

### 2.2.1   Study area

The study area covers the French Alps and Pyrenees. In all operational productions of avalanche hazard forecasting, these

regions are divided respectively in 23 and 11 massifs (Fig. 1) indentical to the ones used in SAFRAN discretization (Sect. 2.1.2). The climate is contrasted, colder and wetter in Northern Alps, much drier in Southern Alps and Eastern Pyrenees due to the Mediterranean influence (Durand et al., 2009). White dots correspond to stations where daily meteorological and snow observations are available in winter in the so-called "nivo-météorologique" observation network.

### 2.2.2   Predictors

Two separate sets of training data for the statistical postprocessing were used in this study as predictors. Their specificies detailed below are also summarized in Table 1.

**Reforecasts used for training**

First, the PEARP reforecasts consist of 10 members including one control member and were issued in 2018. The reforecasts (Boisserie et al., 2016) are based on a homogeneous model configuration identical to the operational release of 5th December

2017 (same resolution and physical parametrizations), but they only include physical perturbations and no perturbation of the initial state, contrary to operational PEARP forecasts. The initial states are build with ERA-Interim reanalysis (Dee et al.,

**Figure 1.** Map of massifs in French Alps (top) and Pyrenees (bottom), with altitude in meters and the observation stations represented as white dots





**Table 1.** Summary of predictors dataset used for training and evaluation

| Predictor | Use | Members | Perturbation of initial conditions | Model physics and resolution | Snow simulations geometry | Period |
|---|---|---|---|---|---|---|
| HN reforecasts | Training | 10 | no | constant | stations (Alps and Pyrénées) | 1994-2016 |
| HN real-time forecasts | Training | 35 | yes | variable | 300 m elevation bands (Alps) | 2014-2017 |
| HN real-time forecasts | Evaluation | 35 | yes | constant | stations (Alps and Pyrénées) | 2017-2018 |

2011) for the atmospheric variables and by the 24h standalone coupled forecasts of the SURFEX/ARPEGE model for the earth parameters. These reforecasts were downscaled with SAFRAN for all stations in the French Alps and Pyrenees where snow observations are available. These downscaled forecasts were used to force the Crocus snowpack model to provide simulated heights of new snow. The training period length is 22 seasons (from 1994 till 2016).

**Real time forecasts used for training**

Second, the real-time forecasts of PEARP consist of 35 members including a control member. In contrast to the reforecasts, model configuration has changed over time and the earlier versions were different from the currently operational version (lower horizontal and vertical resolution, different set of model physics). Both physical perturbations and initial state perturbations are included in the real-time forecasts. These forecasts have experimentally forced the S2M snowpack modelling chain in real-

time since 2014. However, these real-time snow forecasts were only issued for the French Alps massifs at specific elevations of 1200, 1500, 1800, 2100, 2400 and 2700 meters. They are used for training over the 2014-2017 period.

    **Real time forecasts used for verification**

Statistical methods have to be evaluated on datasets independent from the ones used for the calibration. Hence, the 35-member real time forecasts of PEARP-S2M covering the 2017-2018 winter were used as predictors for verification (last line

of Table 1). The version of PEARP is homogeneous over the verification period and identical to the reforecast configuration (resolution and physics), but it also accounts for the initial perturbations. Thus, there are more members than in the reforecasts. These verification forecasts were downscaled and forced the Crocus snowpack model for all the stations available in the snow reforecasts. These verification forecasts were used to evaluate all the training scenarios described in Sect. 2.5.

    **Common features**

Note that in all cases (snow reforecasts and real-time snow forecasts used for training and verification), each snow forecast is initialized by a SURFEX/ISBA-Crocus run forced by SAFRAN analysis (assimilating meteorological observations) from the beginning of the season. For each season, the snow reforecasts and real-time forecasts were issued only for months from November to April. Months outside of this time window were neglected due to insufficient observation data. In this study, we consider only HN snow reforecasts and real-time forecasts at four different lead times (+24h, +48h, +72h and +96h).

**Summary**

These two training datasets correspond to two different approaches to estimate the operational model statistical properties. The real-time forecasts represents the closest version to the operational system but on a short period so that unusual events cannot be taken into account. The reforecasts are a simpler version of the ensemble system which do not contain all sources of

error of the system but over a long *climatological* period. The theoretical version of a reforecast would be the exact reproduction of the operational system over a long period, but it is not currently within any national weather service computy facilities to be performed.

#### 2.2.3 Observations

The observation data used in this study have been collected from a network of stations mainly localised in French ski resorts. These observation-stations are illustrated as white dots in Fig. 1. All the observations have been manually measured by the staff members of these resorts. Measurement of the variable of 24h height of new snow (HN24) is done daily by measuring the fresh snow height on top of a measuring board. After each measurement, the board is cleaned. These observations are usually carried out every morning. They can be compared directly to the snow reforecasts which are available at each station. This yielded a total of 113 stations in French Alps and Pyrenees. However, since the real-time snow forecasts used for training are issued only at specific elevations with a 300 m resolution, the observations were in this case associated with the closest standard elevation level in the simulations when the altitude difference was lower than +/- 100 meters, and were ignored for higher differences. This procedure yielded a total of 47 stations only in French Alps.

### 2.3 Post-processing method

Nonhomogeneous Gaussian Regression (NGR) is one of the most commonly used EMOS methods. NGR was first proposed by Jewson et al. (2004); Gneiting et al. (2005); Wilks and Hamill (2007). In these early applications of nonhomogeneous regression, the predictive (i.e. post-processed) distributions are specified as Gaussian. The mean and variance of the Gaussian distribution are typically modelled with a linear regression model using the raw ensemble mean and variance as predictors. Unlike in ordinary regression-based methods, the dependence of the predictive variance on the ensemble variance in nonhomogeneous regressions allows to exhibit less uncertainty when the ensemble dispersion is small, and more uncertainty when the ensemble dispersion is large (Vannitsem et al., 2018). The regression coefficients can be estimated from the training data by using optimization techniques based on the maximum likelihood or minimum Continuous Ranked Probability Score (CRPS) (Gebetsberger et al., 2018). However, Gaussian predictive distributions are not adequate for certain meteorological predictands such as precipitation. This can be solved by transforming the target predictand and its predictors such that it is approximately normal (Baran and Lerch, 2015, 2016) or using non-Gaussian predictive distributions. Many alternative predictive distributions have been proposed. Messner et al. (2014) applied logistic distribution for modelling square-root transformed wind speeds. Generalized Extreme-Value distributions were used by Lerch and Thorarinsdottir (2013) for forecasting the maximum daily windspeed and by Scheuerer (2014) for precipitation. Scheuerer and Hamill (2015) proposed also a nonhomogeneous regression with gamma distribution. Nonnegative predictands such as precipitation have high probability mass at zero, and thus the use of a transformation comes with a number of problems. To address this, nonhomogeneous regression methods based on truncating and censoring of predictive probabilities have been developed. For example, Thorarinsdottir and Gneiting (2010) applied nonhomogeneous regression approach with zero-truncated Gaussian distributions for wind speed forecasting. A zero-truncated distribution is a distribution where random variable have nonzero probability only for positive values and the





negative values are excluded. Censoring instead allows a probability distribution to represent values falling below a chosen threshold. Commonly, in case of ensemble postprocessing of precipitation forecasts, the censoring threshold is set to zero and any negative probability is assigned to zero, providing a probability spike at zero. Scheuerer (2014) used zero-censored GEV predictive distribution in nonhomogeneous regression model for precipitation. Similar approach with zero-censored shifted-

gamma distribution (CSGD) for nonhomogeneous regression was introduced by Scheuerer and Hamill (2015, 2018) and Baran and Nemoda (2016).

Indeed, as precipitation occurrence/non-occurrence and quantity are modelled together, Scheuerer and Hamill argued for using a continuous distribution that permits negative values and left-censoring it at zero. According to their exploratory data analysis, a predictor variable which often takes small values (e.g. the ensemble-mean precipitation forecast) calls for strongly

right-skewed distribution. But as the magnitude of this predictor variable increases, the skewness becomes smaller. This sort of behaviour can be reproduced to some extent by using gamma distributions. The gamma distributions can be defined by a shape parameter $k$ and a scale parameter $\theta$, which are related to the mean $\mu$ and the standard deviation $\sigma$ of the distribution (Wilks, 2011):

$$k = \frac{\mu^2}{\sigma^2}; \theta = \frac{\sigma^2}{\mu} \tag{1}$$

Scheuerer and Hamill (2015) introduce an additional parameter, the shift $\delta > 0$. The purpose of this parameter is to deal with the nonnegativity of gamma distribution by shifting the CDF of gamma distribution somewhat to the left. Therefore, the CSGD model is defined by:

$$\tilde{G}_{k,\theta,\delta}(y) = \begin{cases} G_k\left(\frac{y-\delta}{\theta}\right) & \text{for } y \geqslant 0 \\ 0 & \text{for } y < 0 \end{cases} \tag{2}$$

where $G_k$ denotes the Cumulated Distribution Function of a gamma distribution with unit scale and shape parameter $k$. This

distribution can be parameterized with $\mu$, $\sigma$ and $\delta$ by using Eq. (1).

In the non homogeneous regression model defined by Scheuerer and Hamill (2018), when the predictability becomes weak, the forecast CSG distribution converges towards a CSG distribution of mean $\mu_{cl}$, standard deviation $\sigma_{cl}$, and shift $\delta_{cl}$, corresponding to the best fit of the gamma law with the climatological distribution of observations. The validity of the adjustment of the climatology with a gamma law was verified over the whole observations dataset in Fig. 2. It only exhibits a small

underestimation of extreme values (for frequencies of exceedance lower than 1/1000).

Thus, for a given day, $\mu$, $\sigma$ and $\delta$ are linked to the raw ensemble forecasts with the regression model of Scheuerer and Hamill (2018):

$$\mu = \frac{\mu_{cl}}{\alpha_1} \log 1p[\exp m1(\alpha_1)(\alpha_2 + \alpha_3 \text{POP} + \alpha_4 \frac{\bar{x}}{\mu_{cl}})] \tag{3}$$


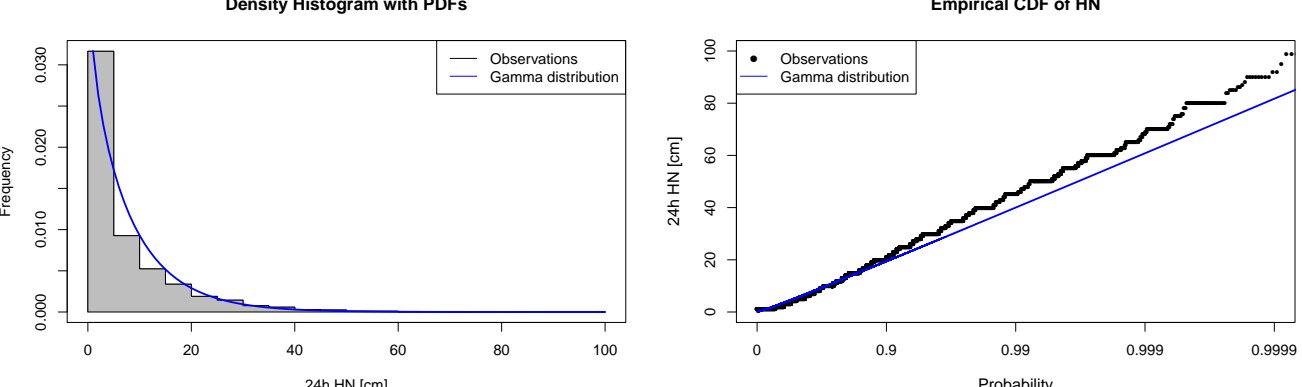

**Figure 2.** Distribution functions of positive HN observations over all dates and stations. Left: Frequency histogram of the raw data (gray) and Probability Density Function of the fit with a gamma law (blue). Right: Cumulative Distribution Functions of observations (black) and gamma law (blue) with a focus on the distribution tail.

$$\sigma = \beta_1 \sigma_{cl} \sqrt{\frac{\mu}{\mu_{cl}}} + \beta_2 \mathrm{MD} \tag{4}$$

$$\delta = \delta_{cl} \tag{5}$$

In Eq. (3), $\mathrm{log1p}(u) = \log(1+u)$ and $\mathrm{expm1}(u) = \exp(u) - 1$. In this regression model, the ensemble forecasts are summarized by the ensemble mean $\bar{x}$, the probability of precipitation POP, and the ensemble mean difference MD (a metric of

5   ensemble spread), as defined by Eq. (6), (7) and (8):

$$\bar{x} = \frac{1}{M} \sum_{m=1}^{M} x_m \tag{6}$$

$$\mathrm{POP} = \frac{1}{M} \mathcal{I}_{x_m > 0} \tag{7}$$

$$\mathrm{MD} = \frac{1}{M^2} \sum_{m=1}^{M} \sum_{m'=1}^{M} |x_m - x_{m'}| \tag{8}$$

with $x_m$ the forecast of each member $m$ among the $M$ members, and $\mathcal{I}_{x_m > 0} = 1$ if $x_m > 0$, 0 otherwise.





The regression coefficients $\alpha_1$, $\alpha_2$, $\alpha_3$, $\alpha_4$, $\beta_1$ and $\beta_2$ are estimated by the optimization process used by Scheuerer and Hamill (2015) as described in next section.

In addition to the convergence of this model towards the climatological distribution for weak predictability, this model also include several advantages compared to standard non-homogeneous regressions. First, the POP predictor can improve the forecast distribution compared to models based only on the ensemble mean by providing complementary information about the expected precipitation occurrence. Then, the links between $\mu$ and $\bar{x}$ and between $\mu$ and POP are not supposed to be linear in Eq. (3) (the model tends to the linear case when $\alpha_1 \to 0$). Finally, Eq. (4) introduces an explicit heteroscedasticty ($\sigma$ does not only depend on MD but also on $\mu$). This important property for precipitation (or snowfall) may not be sufficiently reproduced by the spread of the raw ensemble. Extended justifications of the form of this regression model are provided in Scheuerer and Hamill (2015, 2018).

## 2.4 Evaluation metrics

It is commonly admitted that reliability and resolution are the two main properties to qualify the skill of a probabilistic prediction system (Candille and Talagrand, 2005). Reliability is defined as a statistical consistency between the predicted probabilities and the subsequent observations. For instance, a probabilistic prediction system is reliable if a given snowfall occurs with frequency $p$ when it is predicted to occur with the probability $p\ \forall p \in [0,1]$. A system can be reliable if it would always predict the climatological distribution of the atmospheric variable under consideration. However, that would lack practical usefulness and therefore the second property, resolution, implies that the individual spread of the predicted distributions must be smaller than the climatological spread.

### 2.4.1 CRPS

The Continous Ranked Probability Score (CRPS) is one of the most common probabilistic tool to evaluate the ensemble skill both in terms of reliability (unbiased probabilities) and resolution (ability to separate the probability classes) (Candille and Talagrand, 2005). For a given forecast, the CRPS corresponds to the integrated quadratic distance between the CDF of ensemble forecast and the CDF of observation. Commonly, the CRPS is averaged over $N$ available forecasts following Eq. (9):

$$\overline{\text{CRPS}} = \frac{1}{N} \sum_{i=1}^{N} \int_{\mathbb{R}} (F_i(x) - H(x - o_i))^2 \mathrm{d}x \tag{9}$$

where $F_i(x)$ is the cumulative distribution function of the ensemble simulation at time $i$, $o_i$ the observation at time $i$, and $H(y)$ is the Heaviside function ($H(y) = 0$ if $y \leq 0$; $H(y) = 1$ if $y > 0$). CRPS value has the same unit as the evaluated variable and tends towards 0 for a perfect system. Note that in the case of a CSG distribution (when $F_i = \tilde{G}_{k,\theta,\delta}$), an analytic expression of CRPS allows to directly compute the score from the parameters $k$, $\theta$ and $\delta$ (Scheuerer and Hamill, 2015). In this study, $\overline{\text{CRPS}}$ is used to optimize the 6 regression parameters ($\alpha_1$, $\alpha_2$, $\alpha_3$, $\alpha_4$, $\beta_1$, $\beta_2$) of Eq. (3) and (4), by minimizing this score on the training data. We remind that the correspondence between ($\mu$, $\sigma$) and ($k$, $\theta$) is given by Eq. (1). Then, $\overline{\text{CRPS}}$ is also used to





evaluate the overall skill of the HN raw forecasts of the PEARP-S2M system. Finally, to assess the improvement obtained by the post-processing compared to the raw forecasts, we compute the Continuous Ranked Probability Skill Score:

$$\text{CRPSS} = 1 - \frac{\overline{\text{CRPS}}}{\overline{\text{CRPS}}_{\text{ref}}} \tag{10}$$

where $\overline{\text{CRPS}}_{\text{ref}}$ is the reference mean CRPS of the raw ensemble forecasts. Therefore, positive CRPSS values indicate an
improvement compared to the raw forecasts. In this work, $\overline{\text{CRPS}}$ and CRPSS were computed separately for each station and we present the distribution of these scores among stations.

### 2.4.2   Rank histograms and quantile-quantile plots

Statistical post-processing are mainly expected to improve the reliability of ensemble forecasts systems. Therefore, we chose to present complementary diagnostics to better illustrate the improvement obtained by the postprocessing in terms of reliability
compared to the raw forecasts. For that purpose, we used rank histograms and quantile-quantile plots.

Rank histograms (Hamill, 2001) illustrate the occurrence frequency of the different possible ranks of the observations $o_k$ among the sorted ensemble members. The flatness of this histogram is a condition of the system reliability (if the simulated probabilities are unbiased regardless of the probability level, the different ranks should have a uniform occurrence frequency). It is also an indicator of the spread-skill as underdispersion will result in a U-shaped rank histogram and overdispersion in a bell-
shaped rank histogram. Rank histograms are commonly computed for the whole forecast dataset, but this can hide contrasted behaviours between the different parts of the distribution. Forecast stratification (Broecker, 2008), as the process of dividing the whole dataset into different subsets and computing verification metrics for each subset, has been introduced as a way to better diagnose where the deficiencies of the forecast system lie. Bellier et al. (2017) compared different strategies for the stratification criteria, based on either the observations or the forecasts, and justified the use of a forecast-based stratification criteria for
verification rank histograms. Indeed, they showed that conditionning the rank histogram to observations is likely to draw erroneous conclusions about the real behaviour of ensemble forecasts. Therefore in this study, a forecast-based stratification is used by considering three HN intervals $[0\,\text{cm}, 10\,\text{cm}[, [10\,\text{cm}, 30\,\text{cm}[$ and $[30\,\text{cm}, +\infty[$ for the ensemble median. To guarantee a sufficient sample size for rank histograms, they are computed for the whole evaluation dataset by considering all dates and stations as independent.
To understand the connection between forecast errors and the magnitude of observed HN, quantile-quantile plots of sorted observations as a function of the forecast quantiles for the equivalent frequency levels are also presented. Contrary to the rank histograms, quantile-quantile plots do not discriminate the probability classes in the ensemble forecasts but it allows to verify that the postprocessing removes the biases for any value of the forecast variable, with a reduced constraint on sample size compared to the stratified rank histogram. Similarly to the rank histograms, quantile-quantiles plots are computed for the
whole evaluation dataset (all dates and stations).





## 2.5 Experiments

The post-processing method described in Sect. 2.3 was calibrated on the data listed in Sect. 2.2.2. Several experiments were performed. First, for each station, the predictor is the simulated HN from the snow reforecast, and the predictand is the observed HN at the station. This leads to a different calibration for each station. Then, for each massif, the same predictors and

predictands of all stations inside the massif boundaries are mixed in the same training vectors as independent events. This method will be further referred as massif-scale calibration because it leads to a unique calibration for each massif. The results of these two first experiments are described in Sect. 3.2.1. Finally, the same massif-scale calibration is applied by using the real-time forecasts as predictors. The comparison of both training dataset is analysed in Sect. 3.2.2. The skill of the raw forecast and all postprocessing experiments is assessed with the independent evaluation dataset described in Sect. 2.2.2 with the metrics

of Sect. 2.4.

## 3  Results and discussion

### 3.1  Evaluation of raw forecasts

Raw HN real-time forecasts of winter 2017-2018 (Sect. 2.2.2) are evaluated. The CRPS at different lead times of the raw forecast is given in Fig. 3a. The boxplots represent the variability of the score between stations, which is relatelively large at

all lead times. The mean CRPS slightly deteriorates with longer lead times.

The rank histogram of the raw forecast is presented in Fig. 3b and stratified according to the ensemble mean with three different categories (low subset in blue: 0-10 cm, medium subset in green: 10-30 cm, high subset in red: above 30 cm) . The raw HN forecasts are biased with high underdispersion on all three subsets (U-shape). Above 10 cm forecasts, about 50% of the observed values are not included in the ensemble (rank 1 or rank 36). Note that the small sample size of the high subset (79

events) causes a high sampling variability in the different probability classes.

To understand the link between forecast errors and the magnitude of HN, Q-Q plot of sorted observations as a function of the forecast quantiles for the equivalent probability levels is presented in Fig. 3c. The systematic bias in the forecast increases as the observed HN increases. However, since the sample size of high observed HN is small, we can expect significant sampling variability in the upper tail.



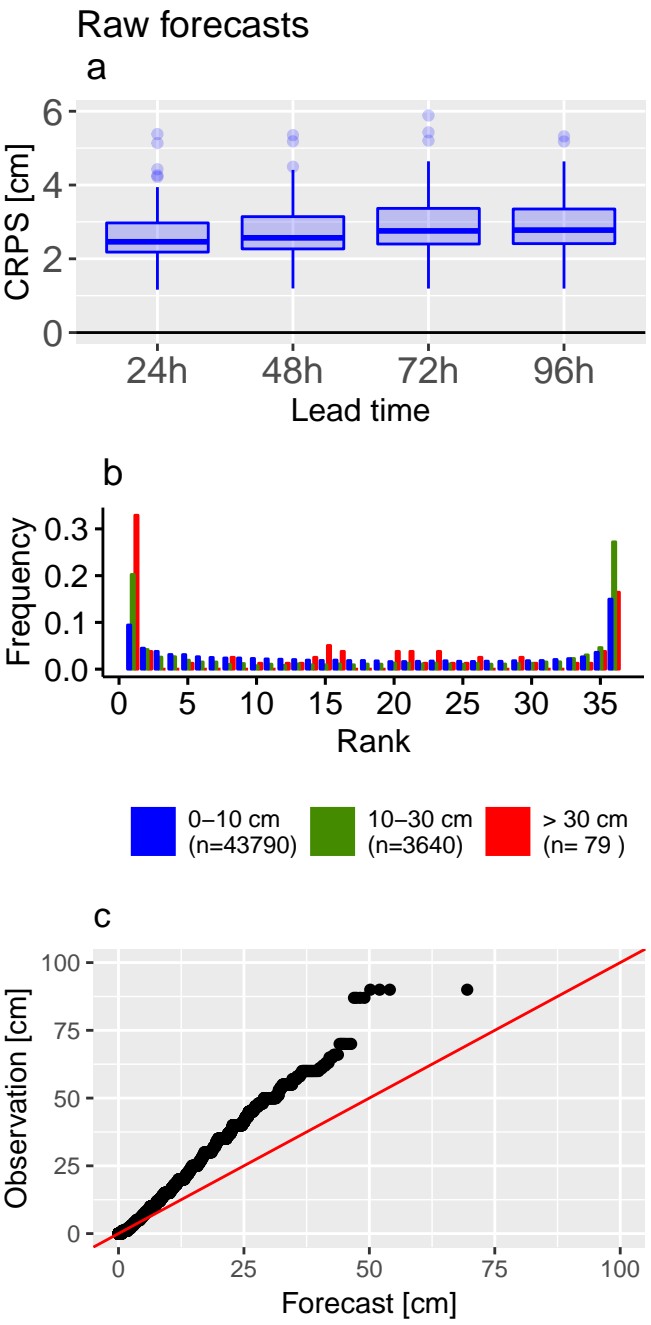

**Figure 3.** Evaluation of raw HN forecasts from PEARP-S2M during winter 2017-2018. a) CRPS of HN as a function of prediction lead time. The boxplot represents the variability of scores between the 113 stations. b) Ranks histograms of HN forecasts for three classes of HN ensemble median (indigo: $[0\,\mathrm{cm}, 10\,\mathrm{cm}[$, cyan: $[10\,\mathrm{cm}, 30\,\mathrm{cm}[$, red: $[30\,\mathrm{cm}, +\infty[$). c) Quantile-Quantile plot: the black dots represent sorted observations as a function of the forecast quantiles for the equivalent frequency levels. Red line illustrates the ideal distribution.





## 3.2 Evaluation of post-processed forecasts

### 3.2.1 Comparison of local-scale and massif-scale training

In this section, we present only the results obtained by using the reforecast dataset as training and we compare the impact of a local-scale calibration for each station to a massif-scale calibration where all observations in the same massif are mixed in the
same training vector in order to obtain only one set of parameters by massif for Eq. (3) and (4), that can be applied at any point in the massifs. The evaluation is performed on 113 stations in 30 different massifs.

CRPSS of each station for the verification period and with the raw forecast as a reference are presented in Fig. 4a and 4b. Postprocessing with both local-scale and massif-scale training significantly improves the CRPS in the majority of the stations (positive skill scores for a large majority of stations), although in both cases, the improvement decreases with longer lead
times. The CRPSS of local-scale training is slightly better than the CRPSS of massif-scale training on smaller lead times (24h and 48h), but for longer lead times (72h and 96h), the difference between local-scale and massif-scale decreases. Overall, the difference between local-scale and massif-scale training according to the CRPSS is limited.

The rank histograms of the postprocessed ensembles with local-scale and massif-scale snow reforecast training are presented in Fig. 4c and 4d. In both cases the shape of the histograms are similar, and show that the reliability has been greatly improved
compared to the raw forecast (Fig. 3b). This is the expected behavior of the postprocessing and obtaining such a result on the validation period independent from the training period proves the robustness of the model. However, a relative overdispersion of the post-processed can be noticed (slight bell-shape of the histograms). As for the rank histogram of the raw forecast, the sample size of the high subset is small and causes variability in the corresponding rank histogram (red bars).

Q-Q plots of both postprocessing training scenarios with the snow reforecast are presented in Fig. 4e and 4f. In both cases,
a significant improvement compared to the raw Q-Q plot (Fig. 3c) can be noted over all quantiles. Indeed, the Q-Q plot shows that the postprocessed forecasts and the observations have almost the same climatological distribution. Again, the sample size of high observed HN is small, and causes sampling variability in the upper tail.

Examples of raw and postprocessed ensembles with the snow reforecast training in local-scale and massif-scale are given in Fig. 5. In all four cases, the CRPSS are around +30%, showing a clear improvement of the forecasts by the postprocessing over
January 2018. Note that the scores over this short period are provided for the example but only the previous scores computed over the whole evaluation period (Fig. 4) should be considered for robust conclusions. The better improvement is obtained with the local scale training in the first example (Fig. 5a and 5b) but with the massif scale training in the second example (Fig. 5c and 5d). These examples show that some differences can be observed between the skill of the local-scale and the massif-scale training but with variability between stations and no systematic improvement or deterioration. This is consistent with the
similar scores presented before between both spatial scales. In both examples and regardless the spatial scale of training, the post-processing increases the median and the spread compared the raw ensemble, consistently with the systematic negative bias and underdispersion of the raw forecast observed in Fig. 3b and 3c. Thus, for most days with observed snowfall, the observations fall inside the EMOS quantiles whereas they frequently fall outside the raw ensemble. Nevertheless, the method





causes overdispersion especially visible by adding spread even for the days when all the raw forecast members are predicting no snowfall.

### 3.2.2 Comparison of real-time snow forecast and snow reforecast training

In this section, we present only the results obtained with a massif-scale training and we compare the impact of using the
reforecast dataset or the real-time forecast dataset as training, which do not have the same advantages and disadvantages. As mentioned in Sect. 2.2.2, only 47 stations in 21 different massifs were included in this comparison. Note that the results obtained in Sect. 3.1 and 3.2.1 are not significantly different between the 113 stations and this subset of 47 stations, as shown by the similar diagnostics obtained for massif-scale reforecast training in both sections (Fig. 4b, 4d and 4f just differ from Fig. 6a, 6c and 6e by the number of stations considered.) CRPSS of each station for the verification period, with the raw forecast as
a reference, are presented in Fig. 6a and 6b. In both cases, the CRPS is improved in the majority of the stations. Similarly to Sect. 3.2.1, the CRPSS is decreasing with longer lead times. However, this decrease is smaller with the real-time snow forecast training and it performs better at 96h lead time compared to the snow reforecast training. As can be noted, the variability of mean CRPSS among the stations is generally higher with the real-time snow forecast training.

The rank histograms of post-processed forecasts for both training scenarios are given in Fig. 6c and 6d. The calibration of
the postprocessed ensembles between these two different training scenarios is different. In case of a snow reforecast training, similar overdispersive behaviour for the low subset (blue) can be noted as was in the previous comparison with a higher number of stations, whereas such issue is not obtained with the real-time snow forecast training. However, the real-time snow forecast training causes a significant positive bias for the medium and high subsets which is not observed with the snow reforecast training.
Q-Q plots with the snow reforecast and the real-time snow forecast are presented in Fig. 6e and 6f. Similar as with the Q-Q plots in the previous comparison, there is a significant improvement over all quantiles compared to the raw Q-Q plot (Fig. 3c). The biases in the ranks from Fig. 6d are not translated in a bias in the simulated quantiles.

Examples for two different stations and lead times of postprocessed ensembles with snow reforecast massif-scale training and the real-time snow forecast massif-scale training are given in Fig. 7. In both examples with the snow reforecast training
(Fig. 7a and 7c), postprocessing increases the spread by stretching the distribution below and above the raw ensemble whereas with the real-time snow forecast training (Fig. 7b and 7d), the distribution is mostly stretched above the raw ensemble. In the example of Fig. 7c and 7d, the real-time snow forecast training performs better since the raw forecast underestimates the snowfall magnitude. However, in the example of Fig. 7a and 7b, postprocessing is improved with the snow reforecast training because the raw forecast overestimates the magnitude of the snowfall and the observations fall multiple times below the raw
forecast. In this example, the postprocessing based on the real-time snow forecast training deteriorates the raw forecast by only stretching the distribution towards higher values. Similarly to the impact of the spatial scale on the training data, there is not any systematic positive or negative impact of the training dataset on the skill of the postprocessing. The main advantages and disadvantages of the real-time snow forecast vs. the snow reforecast training identified in the rank histograms are also emphasized in these examples. First, the overdispersion on dry days obtained by the snow reforecast training can be again





observed in Fig. 7c. This issue disappears with the real-time snow forecast consistently with the satisfactory shape of the low subset (blue bars) in the rank histograms. However, the reliability of the forecasts for severe snowfall events is better with the reforecast training in the example of Fig. 7a and 7b, consistently with the systematic bias of the medium and high subsets (green and red bars) obtained in Fig. 6d.





**Figure 4.** Comparison of postprocessing skill between local-scale training (left column) and massif-scale training (right column) for postprocessed HN forecasts calibrated with the reforecast dataset (1994-2016) and evaluated during winter 2017-2018. a) b) CRPS of HN (cm) as a function of prediction lead time ; the boxplot represents the variability of scores between the 113 stations. c) d) Ranks histograms ; the three HN classes are the same as in Fig. 3b. e) f) Quantile-Quantile plot.

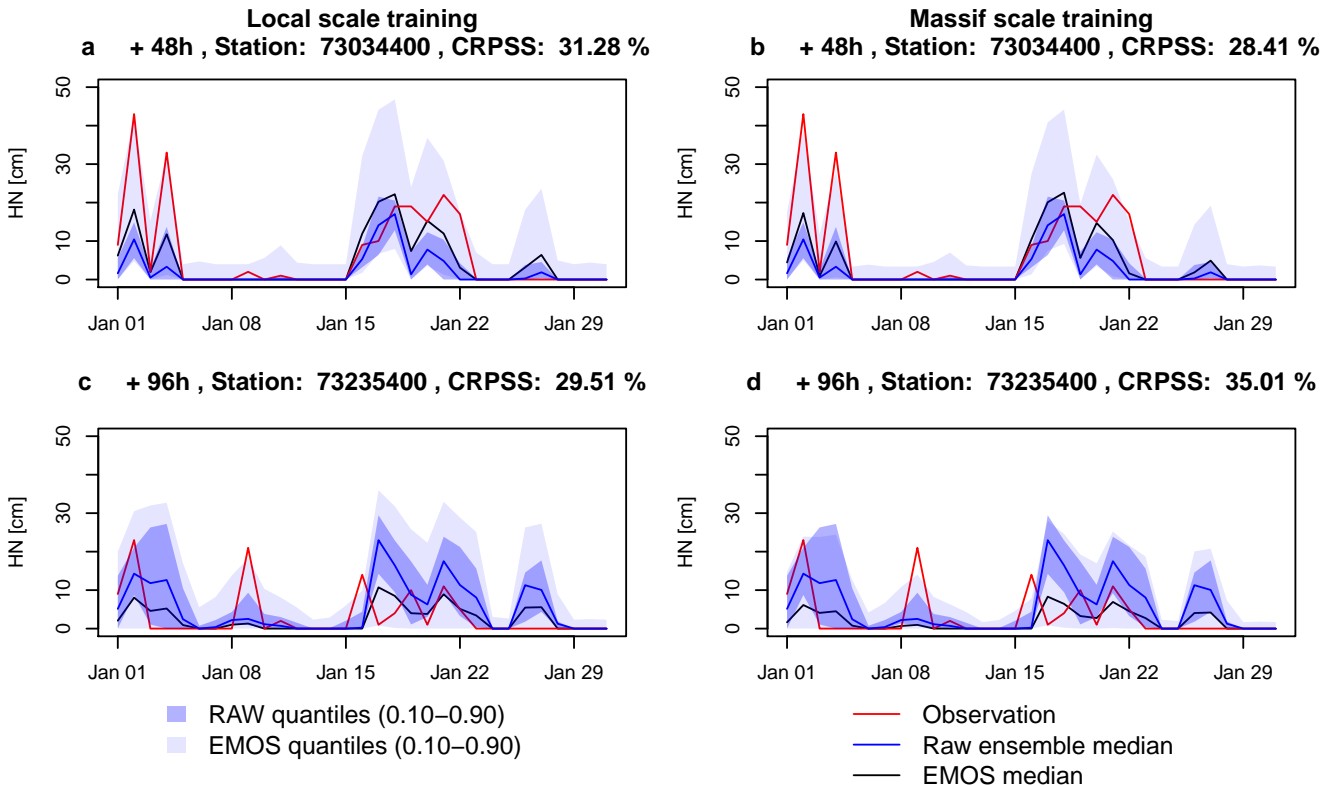

**Figure 5.** Time series of the raw (blue) and postprocessed (gray) ensemble forecasts during January 2018 from a training based on the snow reforecasts. The envelopes represent the interval between the 10th and the 90th percentiles and the solid lines represent the median. These ensemble forecasts are compared to time series of HN observations (red lines). a, b: Example of station 73034400 (Arêches) at the +48h lead time. c, d: Example of station 73235400 (Saint-François-Longchamp) at the +96h lead time. Left column (a,c): Local scale training. Right column (b,d): Massif scale training.

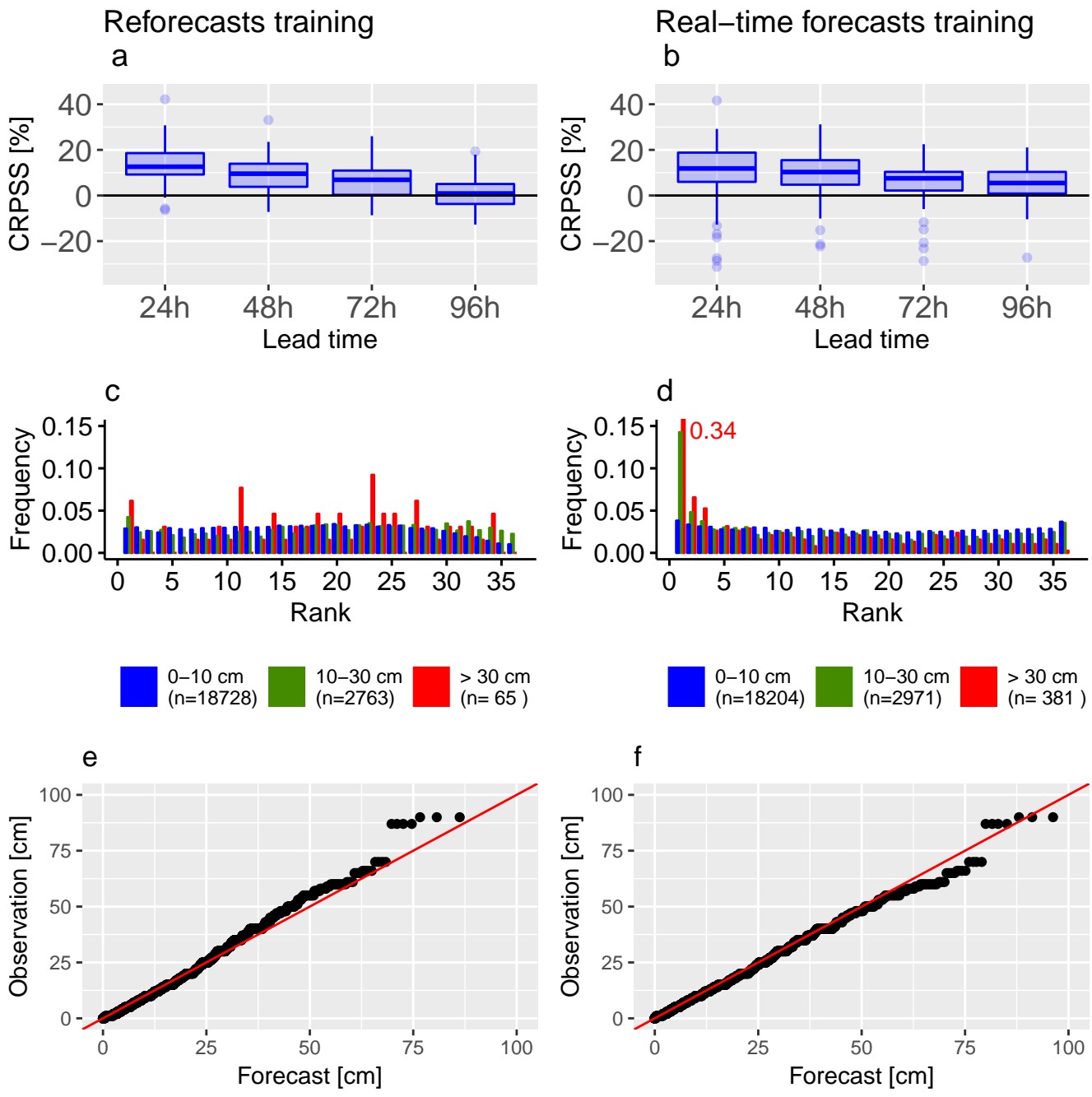

**Figure 6.** Comparison of postprocessing skill between a training with the reforecasts dataset (1994-2016, left column) and a training with the real-time forecasts dataset (2014-2017, right column) for postprocessed HN forecasts calibrated at the massif-scale and evaluated during winter 2017-2018. a) b) CRPS of HN (cm) as a function of prediction lead time ; the boxplot represents the variability of scores between the 47 stations. c) d) Ranks histograms ; the three HN classes are the same as in Fig. 3b. e) f) Quantile-Quantile plot.


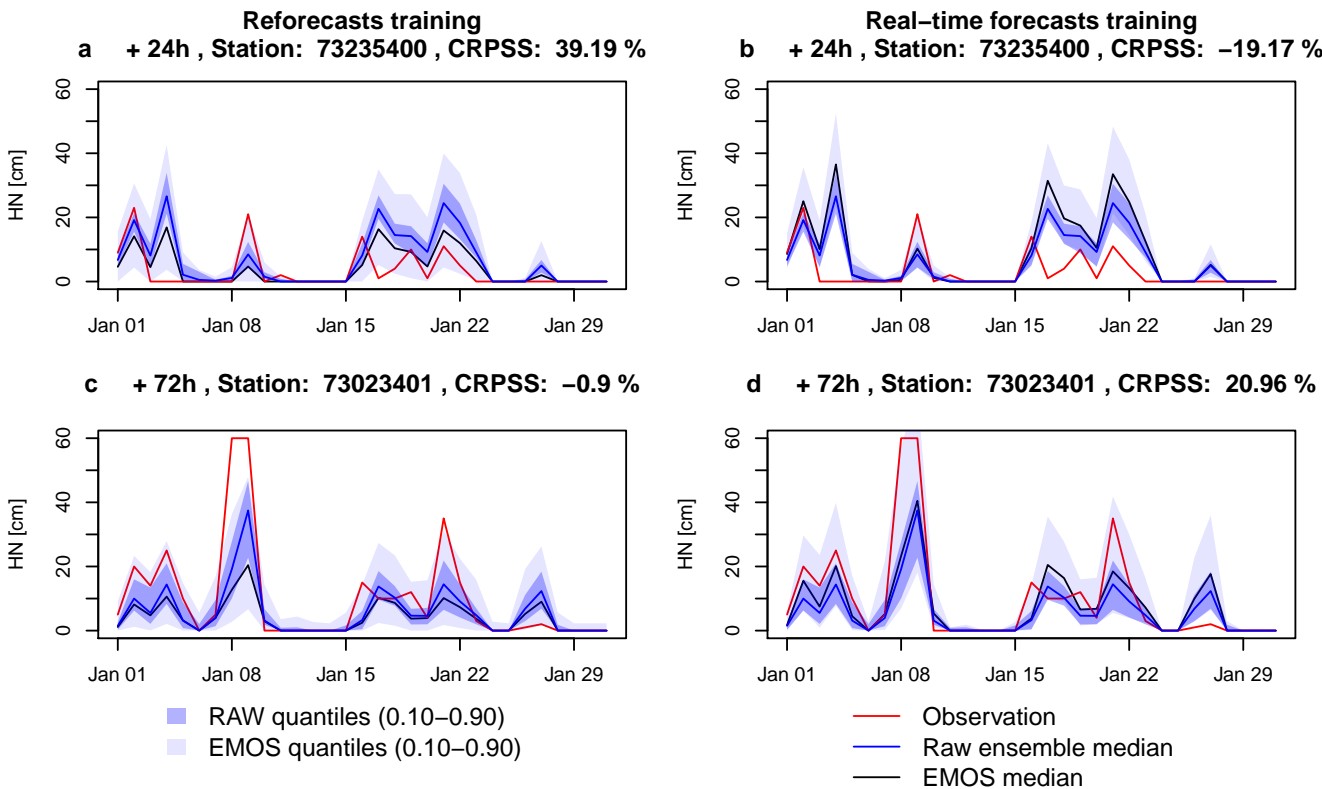

**Figure 7.** Time series of the raw (blue) and postprocessed (gray) ensemble forecasts during January 2018 from a massif-scale training. The envelopes represent the interval between the 10th and the 90th percentiles and the solid lines represent the median. These ensemble forecasts are compared to time series of HN observations (red lines). a, b: Example of station 73235400 (Saint-François-Longchamp) at the +24h lead time. c, d: Example of station 73023401 (Aussois) at the +72h lead time. Left column (a,c): Training with the snow reforecasts. Right column (b,d): Training with the snow real-time forecasts.





## 4   Discussion

### 4.1   Implications for operational automatic forecasts

#### 4.1.1   Added value of postprocessed HN forecasts

Evaluations of the HN raw forecast from the PEARP-S2M ensemble snowpack modelling system in Sect. 3.1 exhibit a signif-
icant underdispersion over all subsets as well as an increasing systematic bias as a function of the height of new snow. It is
the result of a bias and underdispersion of the PEARP precipitation forecasts (Vernay et al., 2015) but also of errors in recent
snow density in the Crocus snowpack model (Sect. 2.1.3) and lack of accounting for uncertainty in the associated processes in
the raw forecasts. Therefore, we recommend to avoid the development of automatic products of HN forecasts based on the raw
simulations.

Statistical processing can help improving the reliability of the forecasts in such products while the correction of these errors
and underdispersion is too challenging to be quickly solved in the NWP and snowpack models. According to the results of
this study, we can state that the use of statistical postprocessing with CSGD method in case of ensemble HN forecasts is
beneficial in most of the evaluated stations in all of the experiments conducted. The extent of these improvements was more or
less similar to what had already been found by several authors in case of statistical postprocessing of ensemble precipitation
forecasts (Gebetsberger et al., 2017; Scheuerer and Hamill, 2015, 2018). However, since statistical postprocessing of ensemble
forecasts had never been applied in the literature on the outputs of a detailed snowpack model, the findings of this study are
very promising in terms of automatic HN forecast developments. Thanks to many advantages of the physical modelling of the
snowpack, the method represents an alternative to the more complex statistical frameworks developed by Stauffer et al. (2018)
and Scheuerer and Hamill (2019) from direct NWP diagnostics as predictors.

### 4.1.2   Spatial scale

Due to the similar improvements whether training data were considered at local-scale or massif-scale (Sect. 3.2.1), the use of
massif-scale training is justified. Indeed, it means that the postprocessing can be applied at any point of the massifs because
homogeneous sets of calibration parameters are obtained for each massif. This is especially interesting for the operational
HN forecasting which has no reason to be limited to observation stations. Note that a potential limitation for applying the
postprocessing anywhere is the relatively limited elevation range of observations used in the evaluations (50% of observation
stations are between 1400 and 2000 m.a.s.l.). Nevertheless, local-scale postprocessing is interesting as well and can be applied
especially if the objective is to gain more reliable HN forecasts for specific locations (e.g. ski resorts).

### 4.1.3   Training dataset

Even though local-scale and massif-scale trainings resulted in similar postprocessing performances, that was not exactly the
case when training forecasts with different lengths and characteristics were compared in Sect. 3.2.2. This comparison resulted
in large differences between the snow reforecast training and the real-time snow forecast training. The reliability of severe

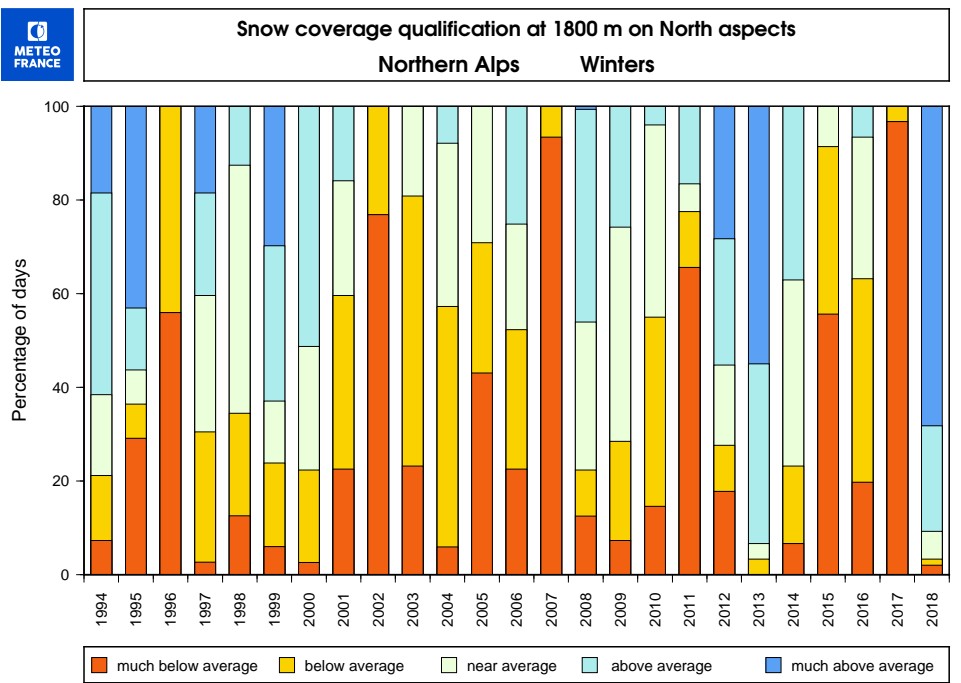

**Figure 8.** Qualification of snow coverage at 1800 m on north aspects in French Northern Alps: for each winter (November-April), percentage of days with total snow depth much below average (lower than 20th climatological percentile), below average (between 20th and 40th percentiles), near average (between 40th and 60th percentiles), above average (between 60th and 80th percentiles), much above average (higher than 80th climatological percentile).

snowfalls was not satisfactory with the real-time snow forecast training. This may be due to the small training length, making highly possible the fact that the climatology of the training period differs from the climatology of the verification period. To understand this issue, Fig. 8 shows the differences in terms of the amount of snow coverage in Northern Alps at 1800 m between the successive seasons. As can be noted, the difference between the snow coverage during the evaluation period

5   (2018) and the training period (2015, 2016 and 2017) is significant. Even when compared to all of the winters in Fig. 8, the year of 2018 is exceptional. Figure 9 presents a rank histogram obtained by cross-validation of three different postprocessing calibrations in which the training periods were 3 years of the 2014-2018 period excluding the evaluation year, repeating this process with 2015, 2016 and 2017 as evaluation year. The rank histogram is the mean of the ranks frequencies of these three simulations. Such cross-validation procedure reduces the impact of seasonal differences in the shape of rank histogram. The

10  shape of the highest subset (red bars) in the cross validated rank histogram of 2014-2017 (Fig. 9) is completely different from the rank histogram obtained for the verification period of 2017-2018 (Fig. 6d). Instead of positive bias, the cross validated verification rank histogram indicates a negative bias. Such behaviour supports the previous arguments about the impact of the

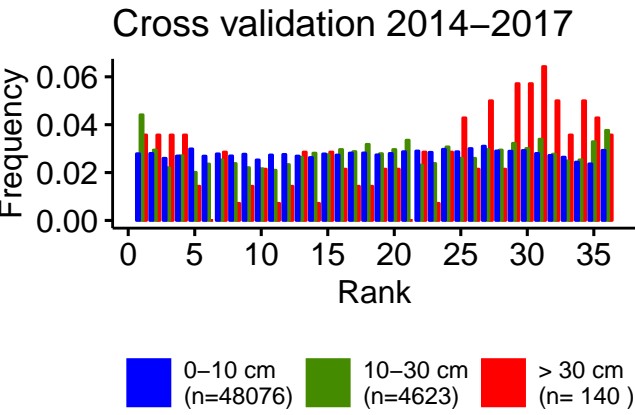

**Figure 9.** Cross-validated ranks histograms of the postprocessed HN forecasts from local-scale calibration with the real-time forecasts dataset. The evaluation is done separately for winters 2014, 2015, 2016 with the training period 2014-2018 excluding the evalution year. The three HN classes are the same as in Fig. 3b.

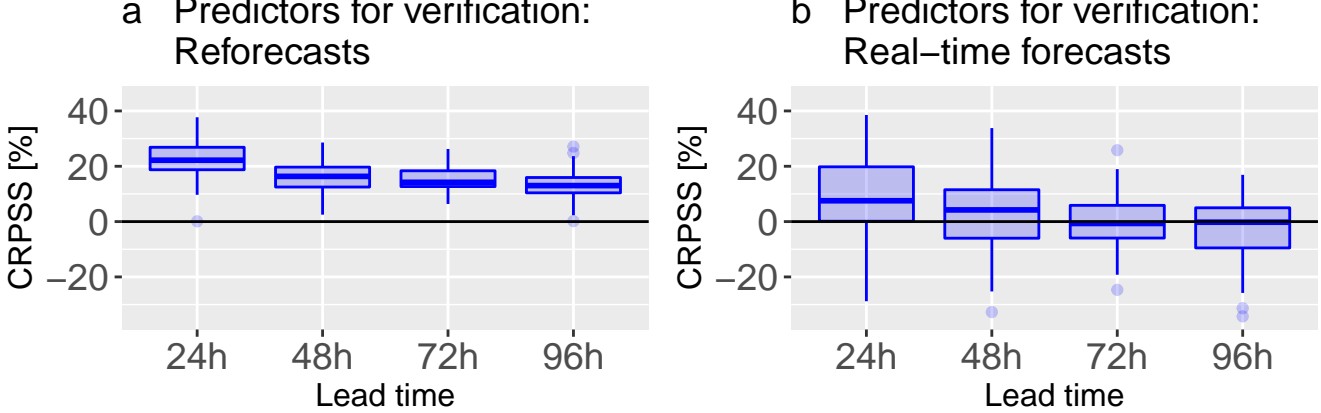

**Figure 10.** CRPS of HN (cm) as a function of prediction lead time for postprocessed forecasts from local-scale calibration with the reforecast dataset (1994-2016) and applied during winter 2015-2016. a: predictors for the verification are taken from the reforecast, b: predictors for the verification are taken from the real-time forecasts. The boxplot represents the variability of scores between the 47 stations in French Alps.

seasonal differences which is especially problematic for operational use in case only short training periods are available for the postprocessing. Indeed, it is highly possible that the upcoming season is significantly different from the past few seasons. Such issue can be avoided or minimized by using longer training periods when reforecasts are available. This conclusion is fully consistent with a significant decrease of the forecast skill obtained by Scheuerer and Hamill (2015) for the highest precipitation

5 amount when reducing the training data length among the same reforecast dataset.



However, the main limitation to the use of the PEARP-S2M reforecast instead of the real time forecasts is the overdispersion generated in the postprocessed forecast. It may be due to the discrepancy in the perturbations and model configurations between these two forecasts. As explained in Sect. 2.2.2, the snow reforecast and the real-time snow forecast have different perturbations and model configurations. The snow reforecast accounts only for the physical perturbations and thus, it has 10 members

whereas the real-time snow forecast accounts for both physical and initial perturbations making it a 35-member forecast. Due to the discrepancy in the perturbations, the real-time snow forecast has higher spread than the snow reforecast. Hence, this can lead to an overcorrection of the spread in case the model is trained with the snow reforecast (lower spread) and verified with the real-time snow forecast (higher spread). To understand the importance of homogeneity between training and verification forecasts, Fig. 10 shows CRPSS comparison between two cases where the postprocessing was applied with the same training

data (the snow reforecast local-scale) and for the same verification period (winter 2015-2016), but first the postprocessing was verified with the snow reforecast as predictor and in the second case the verification was done with the real-time snow forecast as predictor. For this analysis, the verification period had to be included in the training period due to the limited recovery between both datasets. According to the skill scores, the difference between these two forecast evaluations was significant. At all lead times, the skill score with the snow reforecast verification is significantly higher than with the real-time snow forecast

verification. Similar skill scores were also obtained for different seasons when the snow reforecast was used for verification. Hence, the ideal training forecast for statistical postprocessing should be as homogeneous as possible with the verification (operational) forecast. This is an important feedback of this work for research teams in charge of atmospheric modelling: even if the numerical costs are higher, applications of ensemble NWP need reforecasts which include all perturbations implemented in the operational system. Before such dataset is available, it is difficult to choose whether an operational postprocessing should

be based on the snow reforecast or on the real-time snow forecast training as it depends if we prefer to optimize for severe events with high socio-economic impacts or to optimize the spread during dry days (which can be an important factor of confidence for the end-user in an automatic product).

## 4.2 Possible refinements

To maximize the performance of the statistical postprocessing method used in this study, some refinements or extensions could be considered. According to the personal communications with the forecasters of Météo-France, the systematic biases in NWP models may depend on circulation regimes. Hence, categorising the training data by weather types and computing the regression parameters accordingly may be interesting. However, this would decrease the training length and could be problematic especially in the case of the real-time snow forecast training which already had a relatively short training period.

Another extension of the method could be the addition of new predictors. The use of a physical snowpack model reduces

the need of considering both precipitation and temperature variables compared to Stauffer et al. (2018) and Scheuerer and Hamill (2019). Indeed, situations close to the critical threshold of $0^{o}C$ are likely to already result in an increased spread in the raw ensemble forecasts (some members are going to forecast rain, some other to forecast snow). Therefore the post-processed forecasts are naturally exhibiting more spread in this case as it is linked to ensemble spread by Eq. (4). Nevertheless, it is still true than the system may not have the same biases and skill scores depending on various meteorological variables such as





temperature or wind speed, or even depending the month of the year. These variables might be able to improve the statistical relationship in more complex statistical models. Quantile Random Forests for instance could be tested (Taillardat et al., 2016) as they do not need to presume the required predictors in advance and because they could allow combining the different available training datasets by adding a categorical variable. Taillardat et al. (2019) showed that hybrid forest-based procedures produce

the largest skill improvements for forecasting heavy rainfall events over France.

## 5   Conclusion

Various weather services are trying to increase the part of automatic forecasts in their production. This includes the challenging forecast of the height of new snow. The PEARP-S2M modelling system, designed for avalanche hazard forecasting, can also help for this application. Indeed, the PEARP ensemble Numerical Weather Prediction (NWP) model quantifies the uncertainty

of the forecast, the SAFRAN downscaling tool refines the elevation resolution, and the Crocus snowpack model represents the main physical processes responsible for the variability of the height of new snow. However, the raw outputs of PEARP-S2M are biased and underdispersive. The origins of these biases in atmospheric ensembles and snow models are challenging to detect and correct, and hence, a statistical postprocessing of the HN output is necessary. In this study, a nonhomogeneous regression method based on censored shifted gamma distributions was tested to calibrate HN forecasts. The predictands are snow boards

measurements of the height of 24-hour new snow from a network of stations localised in the French Alps and Pyrenees. HN outputs from the PEARP-S2M model chain were statistically postprocessed by considering local-scale and massif-scale training vectors. The method was applied with two different predictor datasets for training (snow reforecast and real-time snow forecast).

    The chosen statistical postprocessing method was found to be successful as the forecast skills were improved for the majority

of the stations in all the conducted experiments. Local-scale and massif-scale trainings had similar improvements and therefore the use of massif-scale training can be preferred for its ability to be applied at a larger spatial scale than the observation points. However, a potential limitation comes with the relatively limited elevation range of the observations since most of the stations are between 1400 and 2000 m.a.s.l. Comparison between the snow reforecast training and the real-time snow forecast training revealed two main challenges. First, due to the higher spread in the verification (operational) forecast than in the snow reforecast

training, the statistical postprocessing ended up overcorrecting the spread. This was found to be especially problematic in case of dry days when it was nearly certain that no snowfall would occur, but still the postprocessed forecast indicated a small probability for snowfall. Second, because of the short training length of the real-time snow forecast, the impact of seasonal differences was found to be significant. In this case, as the training period for the statistical postprocessing was significantly drier than the verification period, the statistical postprocessing did not perform well with higher snowfall events.

An ideal training forecast was identified to be as homogeneous as possible with the operational forecast and to have a long training length. However, such dataset was not available in our case, and before it becomes available, it is difficult to choose if an operational application of postprocessing should be based on the snow reforecast or on the real-time snow forecast since both have advantages and disadvantages. The possibility to initialize an incoming version of PEARP reforecast with an



ensemble of initial states coming for instance from ERA5 reanalyses should be investigated in the future. This should reduce the discrepancy with the operational ensemble system and encourage to prefer postprocessing based on the reforecast than on real-time forecasts. The main limitation remains the high computational time consumption of these reforecasts (Vannitsem et al., 2018) and the balance to find with the frequency of operational changes in NWP and snowpack modelling systems.

*Code and data availability.*   The R code used for postprocessing was originally developed by Michael Scheuerer (Cooperative Institute for Research in Environmental Sciences- University of Colorado Boulder- and NOAA Earth System Research Laboratory, Physical Sciences Division, Boulder- Colorado, USA). The modified version can be provided on request, with the agreement of the original author. The Crocus snowpack model is developed inside the opensource SURFEX project (http://www.umr-cnrm.fr/surfex). The most up-to-date version of the code can be downloaded from the specific branch of the git repository maintained by Centre d'Études de la Neige. For reproductibility of

results, the version used in this work is tagged as "s2m_reanalysis_2018" on the SURFEX git repository (git.umr-cnrm.fr/git/Surfex_Git2. git). The full procedure and documentation to access this git repository can be found at https://opensource.cnrm-game-meteo.fr/projects/ snowtools_git/wiki. The codes of PEARP and SAFRAN are not currently opensource. For reproductibility of results, the PEARP version used in this study is "cy42_peace-op2.18", and the SAFRAN version is tagged as "reforecast_2018" in the private SAFRAN git repository. The raw data of HN forecasts and reforecasts of the PEARP-S2M system can be obtained on request. The HN observations used in this work

are public data available at https://donneespubliques.meteofrance.fr.

*Author contributions.*   BJ developed and run the PEARP reforecast. MV developed and run the SAFRAN downscaling of the PEARP reforecast and real-time forecasts. ML developed and run the SURFEX-Crocus snowpack simulations forced by PEARP-SAFRAN outputs and supervised the study. JPN set up the statistical framework, with scientific contributions of JB and GE. JPN and ML produced the figures and wrote the publication, with contributions of all authors.

*Competing interests.*   The authors declare that they have no conflict of interest.

*Acknowledgements.*   The authors would like to thank Michael Scheuerer for providing the initial code of the EMOS-CSGD, Maxime Taillardat (Météo-France, DirOP/COMPAS) and Marie Dumont (same affiliation as ML and MV) for useful discussions and comments relative to this work, and Frédéric Sassier (Météo-France, DCSC/AVH) for providing Fig. 8. CNRM/CEN, IGE and IRSTEA are part of LabEX OSUG@2020 (ANR10 LABX56).



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
