# Peer review of "Statistical post-processing of ensemble forecasts of the height of new snow"

_Nonlinear Processes in Geophysics, 2019_

## Referee Comment (RC1) · Anonymous Referee #1 · 6 Jun 2019

This manuscript describes the application of a parametric, statistical postprocessing method - originally developed for precipitation amounts - to ensemble forecasts of height of new snow. A number of experiments are performed to explore the benefits and drawbacks of local-scale vs. regional-scale parameter estimation and training with reforecasts or realtime forecasts. The paper is well written and interesting. All conclusions are supported by the presented results. I only have a few minor comments and suggestions to improve language. After these are addressed I recommend the paper for publication with NPG.

Minor comments:

Eq. (3): Is this a typo, or is the ensemble mean indeed divided by mu_cl here? I'm asking because mu_cl is a parameter describing the observation climatology, so it is

not obvious that it can also be used to normalize the forecasts. Please clarify.

p13, l16: So stratification is w.r.t. ensemble mean? Just checking because above you said stratification will be done w.r.t. ensemble median.

p25, l13: I would use the phrase 'was significant' only of statistical tests for significants are performed

Language and typos:

p8, l2: -> is currently not performed within the computing facilities of any national weather service p11, l20: -> tools p12, l20: -> conditioning p12, l27: -> they allow one to p13,l14: -> relatively

---

## Referee Comment (RC2) · Anonymous Referee #2 · 5 Jul 2019

The authors apply ensemble model output statistics (EMOS), which is well established in precipitation forecasting, to new snow height forecasts from the French ensemble snowpack modeling system, which is a novel approach. The authors can also demonstrate skill of the EMOS for their application. An interesting aspect of the manuscript is the comparison of the EMOS on two different scales and the conclusions the authors draw from this comparison (application of the EMOS to not-observed locations). Another strong aspect of the manuscript is the discussion of the limitations of the method in an operational context.

The manuscript is well structured, well written, the experimental setup is suitable, and the results support the conclusions. Therefore, and since it presents a novel approach, the manuscript is suggested for publication.

---

## Author Comment (AC1) · 6 Aug 2019

On behalf of all authors, we thank Anonymous Referee 1 for the value he found in our work and his useful comments and suggestions.

*Minor comments: Eq. (3): Is this a typo, or is the ensemble mean indeed divided by $\mu_{cl}$ here? I'm asking because $\mu_{cl}$ is a parameter describing the observation climatology, so it is not obvious that it can also be used to normalize the forecasts. Please clarify.*

This is a typo in Equation 3. The forecasts are actually normalized by their own climatology $\bar{x}_{cl}$. This normalization and the overall multiplication by the observation climatology $\mu_{cl}$ were found to improve the efficiency of the regression model in the case of very different climatologies between observations and forecasts by Scheurer and

[Figure]

Hamill, 2018. There was not any typo in the code. We corrected the equation in the revised manuscript. Thank you for pointing this out.

*p13, l16: So stratification is w.r.t. ensemble mean? Just checking because above you said stratification will be done w.r.t. ensemble median.*

Indeed, the stratification was based on the ensemble mean in our work, following Bellier et al., 2017. The references to the median in the rank histogram description Page 12 and in the caption of Figure 3 were incorrect and replaced by the mean in the revised manuscript. Thank you for pointing this out.

*p25, l13: I would use the phrase 'was significant' only of statistical tests for significants are performed*

We agree and removed this sentence. We modified the description of this result by a quantitative statement: "At all lead times, the skill score with the snow reforecast verification is higher than with the real-time snow forecast verification: the improvement of the median CRPSS is about 0.12."

*Language and typos: p8, l2: -> is currently not performed within the computing facilities of any national weather service p11, l20: -> tools p12, l20: -> conditioning p12, l27: -> they allow one to p13,l14: -> relatively*

Thank you for pointing out these typos which were all corrected in the revised manuscript.

---

## Author Comment (AC2) · 6 Aug 2019

On behalf of all authors, we thank Anonymous Referee #2 for the value he found in our work and his positive comments.

---

## Author Response (AR2)

On behalf of all authors, we thank the editor Stephan Hemri for the value he found in our work and his careful check of the manuscript.

The mentioned typos have all been corrected in the revised manuscript:

*p3/l11: refered --> referred*
*p5/l31: are built*
*p7/l27: represents -> represent*
*p12,l22: a space is missing: 10cm[, [10cm*
*p25, l34: true than -> true that*

*p5/l4: what is ISBA?*

We added the acronym of ISBA model in the revised manuscript (Interactions between the Soil Biosphere and Atmosphere)

*Figure 1: the color scale for altitude range 2400-3000 and 4200-4800 cannot be distinguished. Probably not relevant, but still not nice.*

We improved the color scale in the revised manuscript to better distinguish all the elevation classes. Note however that the area above 4200 m only covers a few km².

Finally, note that all the typos mentioned by Referee 1 were corrected in our previous submission (6[th] August), as it can be seen in our response to Referee 1. The attached pdf highlights only the changes relative to this previous version.

Best regards

Matthieu Lafaysse, on behalf of all authors

[revised manuscript text omitted]

---

## Author Response (AR3)

*Being a bit picky: I think Referee 1 pointed at both occurrences of the word "significant" on page 25 / line 13. In line with Referee 1 I recommend to replace the phrase "was significant" by something like "was considerable", as you haven't performed any statistical test for this.*

Dear editor,

We apologize for this incomplete correction of Referee 1 remark and applied the recommended correction in our revised manuscript.

Best regards

Matthieu Lafaysse, on behalf of all authors

[revised manuscript text omitted]